# Impact of knee marker misplacement on gait kinematics of children with cerebral palsy using the Conventional Gait Model—A sensitivity study

**Mickael Fonseca**[1,2]*, **Xavier Gasparutto**[1], **Fabien Leboeuf**[3,4], **Raphaël Dumas**[2], **Stéphane Armand**[1]

**1** Laboratory of Kinesiology Willy Taillard, Geneva University Hospitals and University of Geneva, Geneva, Switzerland, **2** IFSTTAR, LBMC UMR_T9406, LBMC, Univ Lyon, Université Claude Bernard Lyon 1, Lyon, France, **3** School of Health & Society, The University of Salford, Salford, United Kingdom, **4** Rehabilitation service, PHU10, Nantes Hospital, Nantes, France

* Mickael.CardosoFonseca@hcuge.ch

**Data Availability Statement:** Data are available on Yareta (DOI: 10.26037/yareta: d7oanpxk3jd2xhd7sg7qhicdv4) and code is

## Abstract

Clinical gait analysis is widely used in clinical routine to assess the function of patients with motor disorders. The proper assessment of the patient's function relies greatly on the repeatability between the measurements. Marker misplacement has been reported as the largest source of variability between measurements and its impact on kinematics is not fully understood. Thus, the purpose of this study was: 1) to evaluate the impact of the misplacement of the lateral femoral epicondyle marker on lower limb kinematics, and 2) evaluate if such impact can be predicted. The kinematic data of 10 children with cerebral palsy and 10 aged-match typical developing children were included. The lateral femoral epicondyle marker was virtually misplaced around its measured position at different magnitudes and directions. The outcome to represent the impact of each marker misplacement on the lower limb was the root mean square deviations between the resultant kinematics from each simulated misplacement and the originally calculated kinematics. Correlation and regression equations were estimated between the root mean square deviation and the magnitude of the misplacement expressed in percentage of leg length. Results indicated that the lower-limb kinematics is highly sensitive to the lateral femoral epicondyle marker misplacement in the anterior-posterior direction. The joint angles most impacted by the anterior-posterior misplacement were the hip internal-external rotation (5.3˚ per 10 mm), the ankle internal-external rotation (4.4˚ per 10 mm) and the knee flexion-extension (4.2˚ per 10 mm). Finally, it was observed that the lower the leg length, the higher the impact of misplacement on kinematics. This impact was predicted by regression equations using the magnitude of misplacement expressed in percentage of leg length. An error below 5˚ on all joints requires a marker placement repeatability under 1.2% of the leg length. In conclusion, the placement of the lateral femoral epicondyle marker in the antero-posterior direction plays a crucial role on the reliability of gait measurements with the Conventional Gait Model.

available from GitLab at https://gitlab.unige.ch/KLab/mis_markers_misplacement.

**Funding:** MF, CRSII5_177179, Swiss National Fond, http://www.snf.ch/en/Pages/default.aspx, The funders had no role in study design, data collection and analysis, decision to publish, or preparation of the manuscript. XG, The author(s) received no specific funding for this work. FL, The author(s) received no specific funding for this work. RD, The author(s) received no specific funding for this work. SA, CRSII5_177179, Swiss National Fond, http://www.snf.ch/en/Pages/default.aspx, The funders had no role in study design, data collection and analysis, decision to publish, or preparation of the manuscript.

**Competing interests:** The authors have declared that no competing interests exist.

## Introduction

Clinical gait analysis (CGA) is widely used in the clinical field to assess functionality of the lower limbs in patients with motor disabilities such as cerebral palsy (CP) [1]. This pathology is considered as the most frequent cause of motor disabilities among children and CGA has been demonstrated to play an important role in supporting decision making for treatment recommendations [2]. In this examination, a set of skin mounted reflective markers is taped on specific bony landmarks on the skin. The three dimensional coordinates of those markers are used to estimate the movement of the bones which constitutes the kinematic outcomes of gait analysis. An accurate assessment is essential to obtain appropriate outcomes and to make a sound decision for treatments. Thus, data arising from motion analysis needs to be reliable [3]. The variability observed in gait data is a consequence of several factors including the measurement system [4], the soft tissue artefacts [5], [6] and the marker placement [7]. The latter has been identified as one of the largest sources of variability in gait analysis and leads to an improper reconstruction of the local coordinate systems used to compute the outcomes of the assessment [8]. More specifically, an average inter-therapist variability up to 9.8 mm on the anterior-posterior (AP) direction was found for the placement of the lateral femoral epicondyle (KNE) among two adults [9]. Finally, to our knowledge, there is no published evaluation of uncertainty in marker placement in the population of children,whether they have CP or are typically developing.

Several reliability studies focused on evaluating the variability induced by marker misplacement over different conditions. Most of the studies [3], [10], [11] reported average kinematic errors of 6° (but up to 25°) in some peak angles among different testers which overcomes the 5° considered as the limit of acceptability by the gait analysis community [3]. In addition, a few studies performed over different laboratories observed values of variability reaching up to 34° for hip rotation [8]. However, the lack of standardized marker placement protocol and anatomical reconstruction was considered as the main factor for the high variability encountered between laboratories [8].

The Conventional Gait Model (CGM) [12] is one of the most used biomechanical models in clinical practice [13]. This model defines the lower limbs through a set of seven segments (pelvis, thighs, shanks, and feet). It is characterised by the computation of the kinematic output through a hierarchical top-down process [14]. Specifically, the KNE marker is involved in the construction of both thigh and shank longitudinal axis as it is used to define respectively the endpoint and origin of these two segments, i.e. the knee joint centre. It is also used to define the medial-lateral axis of the thigh. Thus, misplacement of KNE marker directly influences the hip, knee and ankle angles with the CGM settings.

Despite broad application in clinics, little has been published about the sensitivity of CGM to marker misplacement. Thus, understanding the influence of marker misplacement on the kinematics is critical to obtain a proper interpretation of gait data. Thereby, a few authors studied the influence of marker misplacement on kinematics in adult populations, either by systematically changing a marker position between sessions [15], [16] or by retrospectively simulating a virtual misplacement [17], [18]. All those studies with exception to one [18] relied on the CGM. Kadaba et al. [17] evaluated the sensitivity of knee joint angles to the definition of the femoral mediolateral axis of the CGM on a group of 40 asymptomatic young adults. The authors analytically altered the orientation of this axis up to ±15° and observed that knee adduction-abduction and internal-external rotation were considerably affected (up to 15°). In an experimental study using CGM, Szczerbik and Kalinowska [16] applied systematically a bidirectional shift of 14 mm in the anterior-posterior and proximal-distal direction on the KNE marker to an adult with knee hyperextension and a sixteen year old girl with CP. Their

results showed a maximum of 25˚ of variation on joint angles, with more incidence on the internal-external rotation for the overall joint angles. Finally, a sensitivity analysis by Baker et al. [14] performed on the CGM showed errors up to -1.8˚ of hip internal rotation and 2.2˚ of knee flexion for a misplacement of the KNE marker 5 mm on the anterior and proximal direction. All other angles showed a variation lower than 0.4˚, however, no information was provided regarding the methodology of their analysis. Using an alternative model based on the marker set reported by Pohl et al. [19], Osis et al. [17] simulated a misplacement of several markers independently over a population of 411 adults with common running injuries. They observed errors up to 5.1˚ and 0.6˚ each 10 mm of KNE marker misplacement on the AP and proximal-distal (PD) directions respectively.

To better understand the relationship between KNE marker misplacement and kinematics, it is important to perform a complete analysis by considering the concomitant effect of direction and magnitude of marker misplacement with populations encountered in clinical practice. To our knowledge, no study using specifically the CGM has considered the impact of KNE misplacement combining both multiple directions and magnitudes of misplacement. Moreover, most of these studies were made with healthy adults. Due to skeletal deformities and altered gait patterns, the results from the studies performed on adult subjects cannot be generalised to children, especially with CP. It can also be anticipated (according to trigonometric rules and differences in size) that the errors on the kinematics would be amplified in children for a similar misplacement error. Thus, the aims of the present study were: 1) to evaluate the sensitivity of the CGM model to marker misplacement of the lateral epicondyle marker within children with CP and 2) to identify a potential correlation between root mean square deviation (RMSD) error and misplacement magnitude normalized by the anthropometric parameters of the patients.

## Material and methods

### Data collection

This study was approved by the "Commission Cantonale d'Éthique de la Recherche" of Geneva (CCER-2018-00229) and all participants provided written informed consent. Gait data of ten children (6 males and 4 females, mean (standard deviations), [range]: 12.4 (4.7), [6–18] years; height: 150.0 (22.7), [119–187.5] cm and weight: 45.1 (26.4) [14.8–106] kg) with CP (GMFCS level I-II; five bilateral and five unilateral) acquired during clinical routine was retrospectively included. The height of the participants presented, intentionally, a large range with a stepwise increase of 7.7 ± 2.0 cm (mean ± standard deviations) between participants. Ten typically developing children (8 males and 2 females, mean (standard deviations), [range]: 13.7 (3.16), [9–18] years; height: 160.8 (19.1), [127–191] cm and weight: 49.5 (17.7), [24–83] kg were included as a control group. One experienced examiner with more than 10 years of regular practice in marker placement and that follows guidelines for marker placement and anatomical palpation [20] has performed the marker placement.

### Testing procedure

The workflow of the study is represented in Fig 1. The anthropometric data (leg length, knee and ankle width, height and weight) were collected by experienced physiotherapists. Then, participants were equipped according to the CGM marker set [12] (12.5mm) and asked to walk barefoot at a self-selected speed. A 12-camera motion capture system (Oqus7+, Qualisys, Göteborg, Sweden) tracked the marker trajectories at 100Hz. Gait kinematic was processed with the Vicon PiG clone, provided as CGM1.1 by the open-source library PyCGM2, which requires a static trial for calibration [21]. In agreement with the original version of the CGM

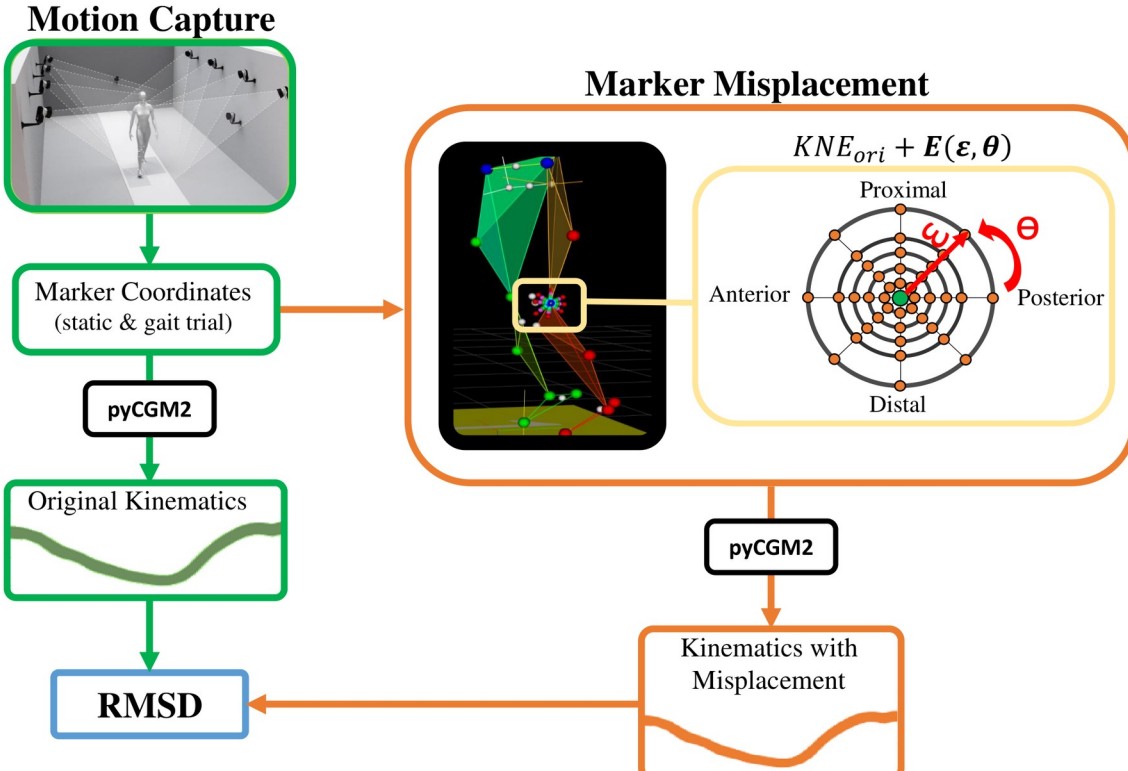

**Fig 1. Workflow for sensitivity analysis.** From gait measurement, one static and one gait trial were considered. The original marker set was used to calculate the reference kinematics (green path). Coordinates of KNE marker were systematically misplaced as a function of angle direction (θ) and magnitude (ε) and the kinematics was calculated for each misplacement (orange path). Finally, the RMSD was calculated as a function of each misplacement kinematics (Err$_i$) and the reference kinematics (O$_i$) (grey).

[14], the coronal plane of the femur was constructed from the hip joint centre, the KNE and the lateral thigh wand mounted marker. A systematic offset was introduced to the KNE marker along the AP and PD axis of the thigh to create a virtual marker. Virtual markers were placed every 45˚ around the original position at five different magnitudes (distance from the original marker): 5, 10, 15, 20 and 30 mm leading to a total of 40 virtual misplacements of the KNE position for each patient. The Eq (1) estimates the new position of the KNE marker ($KNE_{misp}$) as a result of the sum of the original position ($KNE_{ori}$) on the segment coordinate system with an error ($E$) defined in function of magnitude ($\varepsilon$) and direction ($\theta$).

$$KNE_{misp} = KNE_{ori} + E(\varepsilon, \theta) \tag{1}$$

## Statistical analysis

The quantification of the influence for each misplacement on the kinematics was assessed by the RMSD between the originally calculated angle value and the angle resultant from the misplaced KNE marker (2):

$$RMSD = \sqrt{\frac{\sum_{i=1}^{n}\left(Err_i - O_i\right)^2}{n}} \tag{2}$$

where $n$ is the number of frames, $O_i$ the original angle and $Err_i$ the angle resultant from marker misplacement. Discrete and continuous parameters of the gait cycle were considered.

Values of error under 2˚ were considered as within the optimal value, errors between 2˚ and 5˚ were considered within the acceptable interval and errors above 5˚ were considered too high for clinical interpretation as previously reported [3]. Finally, the coefficient of correlation between the RMSD with the misplacement magnitude normalized by leg length and knee width were estimated by a Pearson correlation. Altman's guidelines were used to interpret the correlation as: poor if R≤0.2, fair if 0.2< R ≤0.4, moderate if 0.4< R ≤0.6, good if 0.6< R ≤0.8, and very good, if R >0.8 [22]. Slope (m) and y-intercept (b) were also calculated for each regression line between RMSD error and percentage of leg length (see S1 Table).

## Results

Fig 2 and Table 1 report the RMSD of the kinematics according to each of the misplacements simulated for the whole group of patients. Fig 2 shows that the most affected joint angles are the ankle and hip internal-external rotation and knee flexion-extension as a result from a misplacement on the AP direction. Table 1 provides information relative to the errors resulting from a 10mm misplacement of the KNE marker for both CP and TD group. In the AP direction, hip internal-external rotation RMSD (5.5˚) is above 5˚ while for a misplacement in the

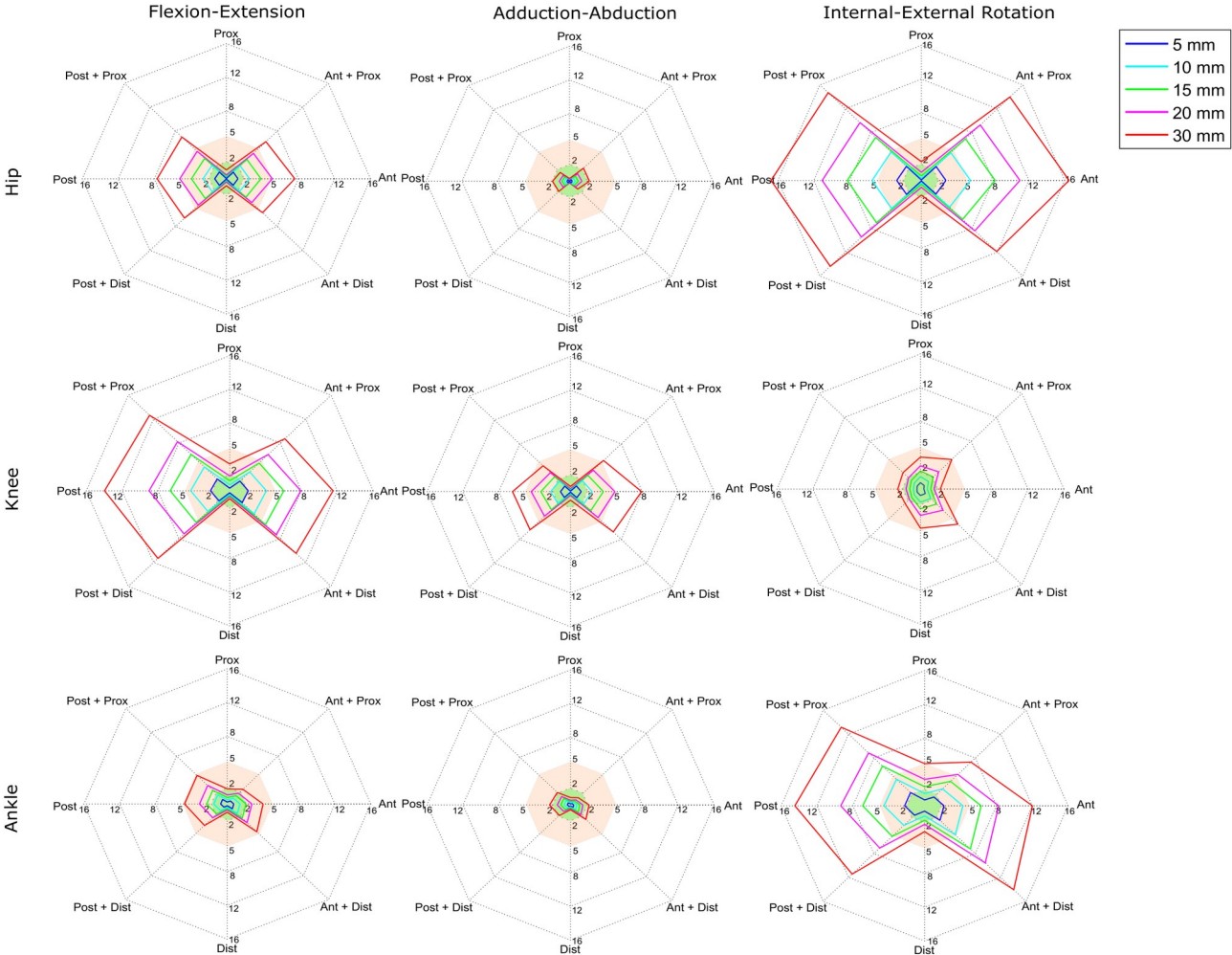

**Fig 2. Impact of KNE marker misplacement on kinematics.** Polar plot representing mean RMSD between marker misplacement and the original position of the overall population of participants respective to each magnitude and direction of misplacement. Green and orange area represents the thresholds of <2˚ (Optimal) and <5˚ (Acceptable) respectively.

**Table 1. RMSD (standard deviations) representing kinematic impact after 10 mm KNE marker misplacement.** Values of RMSD represents the mean difference observed among both populations considering the complete gait cycle and for the CP group considering. Bold fonts represent the values of RMSD over 2˚. RMSD differences between CP and control group (TD) evaluated by the p-value for AP and PD directions.

| Joint | Angle | Anterior | | Posterior | | Proximal | | Distal | |
|---|---|---|---|---|---|---|---|---|---|
| | | CP | TD | CP | TD | CP | TD | CP | TD |
| **Hip** | Flexion peak | **2.54 (0.5)*** | 2.29 (0.1) | **2.58 (0.6)*** | 2.29 (0.1) | 0.11 (0.1)* | 0.04 (0.0) | 0.10 (0.0)* | 0.04 (0.0) |
| | Flexion-Extension GC | **2.57 (0.5)*** | **2.26 (0.1)** | **2.59 (0.6)*** | **2.25 (0.1)** | 0.11 (0.1)* | 0.04 (0.0) | 0.10 (0.0)* | 0.04 (0.0) |
| | Adduction peak | 0.63 (0.4) | 0.4 (0.2) | 0.55 (0.4) | 0.38 (0.2) | 0.03 (0.0) | 0.01 (0.0) | 0.02 (0.0) | 0.01 (0.0) |
| | Adduction-Abduction GC | 0.63 (0.4) | 0.4 (0.2) | 0.56 (0.3) | 0.39 (0.2) | 0.03 (0.0) | 0.01 (0.0) | 0.02 (0.0) | 0.01 (0.0) |
| | External Rotation peak | **5.19 (1.4)** | 5.45 (0.2) | **5.21 (1.3)** | 5.44 (0.2) | 0.22 (0.1) | 0.09 (0.0) | 0.20 (0.1) | 0.09 (0.0) |
| | Internal-External Rotation GC | **5.46 (1.3)** | **5.61 (0.2)** | **5.48 (1.3)** | **5.55 (0.2)** | 0.23 (0.1) | 0.1 (0.0) | 0.21 (0.2) | 0.09 (0.0) |
| **Knee** | Flexion peak | **3.39 (0.4)** | 3.04 (0.3) | **3.73 (0.5)** | 3.39 (0.4) | 1.31 (0.3) | 0.81 (0.4) | 1.03 (0.3) | 0.77 (0.3) |
| | Flexion-Extension GC | **4.07 (0.7)** | **3.81 (0.4)** | **4.33 (0.8)** | **4.02 (0.4)** | 0.72 (0.2) | 0.54 (0.3)) | 0.42 (0.1) | 0.55 (0.3) |
| | Adduction peak | **3.39 (1.3)** | 3.51 (1.6) | **3.55 (1.4)** | 3.23 (1.5) | 0.12 (0.1) | 0.10 (0.1) | 0.33 (0.2) | 0.11 (0.1) |
| | Adduction -Abduction GC | **2.42 (0.5)** | **2.52 (1.4)** | **2.23 (0.5)** | 1.97 (1.5) | 0.14 (0.1) | 0.10 (0.1) | 0.15 (0.1) | 0.1 (0.1) |
| | External-Rotation peak | 0.60 (0.7) | 0.42 (0.2) | 0.62 (0.8) | 0.42 (0.2) | **2.21 (0.9)** | **2.20 (0.9)** | **2.16 (0.8)** | **2.19 (0.9)** |
| | Internal-External Rotation GC | 0.74 (0.6) | 0.46 (0.2) | 0.77 (0.7) | 0.49 (0.2) | 1.49 (0.4) | 1.49 (0.9) | 1.59 (0.4) | 1.55 (0.9) |
| **Ankle** | Flexion peak | 0.87 (0.3) | 0.81 (0.3) | 1.10 (0.2) | 0.91 (0.3) | 0.51 (0.3) | 0.41 (0.2) | 0.43 (0.3) | 0.41 (0.2) |
| | Flexion-Extension GC | 1.39 (0.3) | 1.18 (0.4) | 1.49 (0.4) | 1.22 (0.3) | 0.49 (0.2) | 0.37 (0.2) | 0.40 (0.2) | 0.38 (0.2) |
| | Internal-External Rotation GC | **4.25 (0.5)** | **3.86 (0.9)** | **4.51 (0.5)** | **4.05 (0.8)** | 1.03 (0.2) | 1.26 (0.7) | 1.23 (0.3) | 1.32 (0.8) |

Values with (*) indicates significant difference between the two groups (p-value < 0.05). GC: Gait cycle, TD: Typically developed.

PD direction no joint angles were above this value. The lowest error for a misplacement in the AP direction was observed on the hip and ankle adduction-abduction with a RMSD up to 2.1˚ for a magnitude of misplacement of 30mm. RMSD differences between CP and TD group were not statistically significant with an exception for the hip flexion-extension. Contrarily, the largest RMSD for the same magnitude, on the AP direction were observed for the hip and ankle internal-external rotation and knee flexion-extension angles with RMSD up to 16.4˚, 14.1˚, 13.5˚, respectively. Misplacement in the PD direction induced considerably less impact than in the AP direction except for the knee angle in the transversal plane (up to 6.5˚).

Offsets were observed between the curves representing different magnitudes of misplacement among the same directions for most of the joint angles (Fig 3). However, the knee adduction-abduction angle showed more a change in amplitude than offset as a function of the KNE marker misplaced in the AP direction. Indeed, the differences were mainly in peak values between the curves (i.e. cross-talk phenomenon).

The correlation calculated between RMSD and misplacement magnitude expressed in percentage of leg length (Table 2) revealed a very good relationship (R > 0.9) for most of the angles with an exception for the ankle adduction-abduction and the knee internal-external rotation for which the correlations were moderated (R<0.61) through AP direction.

## Discussion

The objectives of this study were 1) to evaluate the sensitivity of the CGM model to marker misplacement of the lateral epicondyle marker within children with CP and 2) to identify a potential correlation between the error on kinematics and the misplacement magnitude expressed in percentage of anthropometric parameters of the patients. Our results demonstrated that the largest error on the kinematics introduced by the KNE marker when misplaced in the AP direction (Fig 2), which is in agreement with previous studies [9], [14], [16], [23]. The most altered angles were the hip internal-external rotation, the ankle internal-external

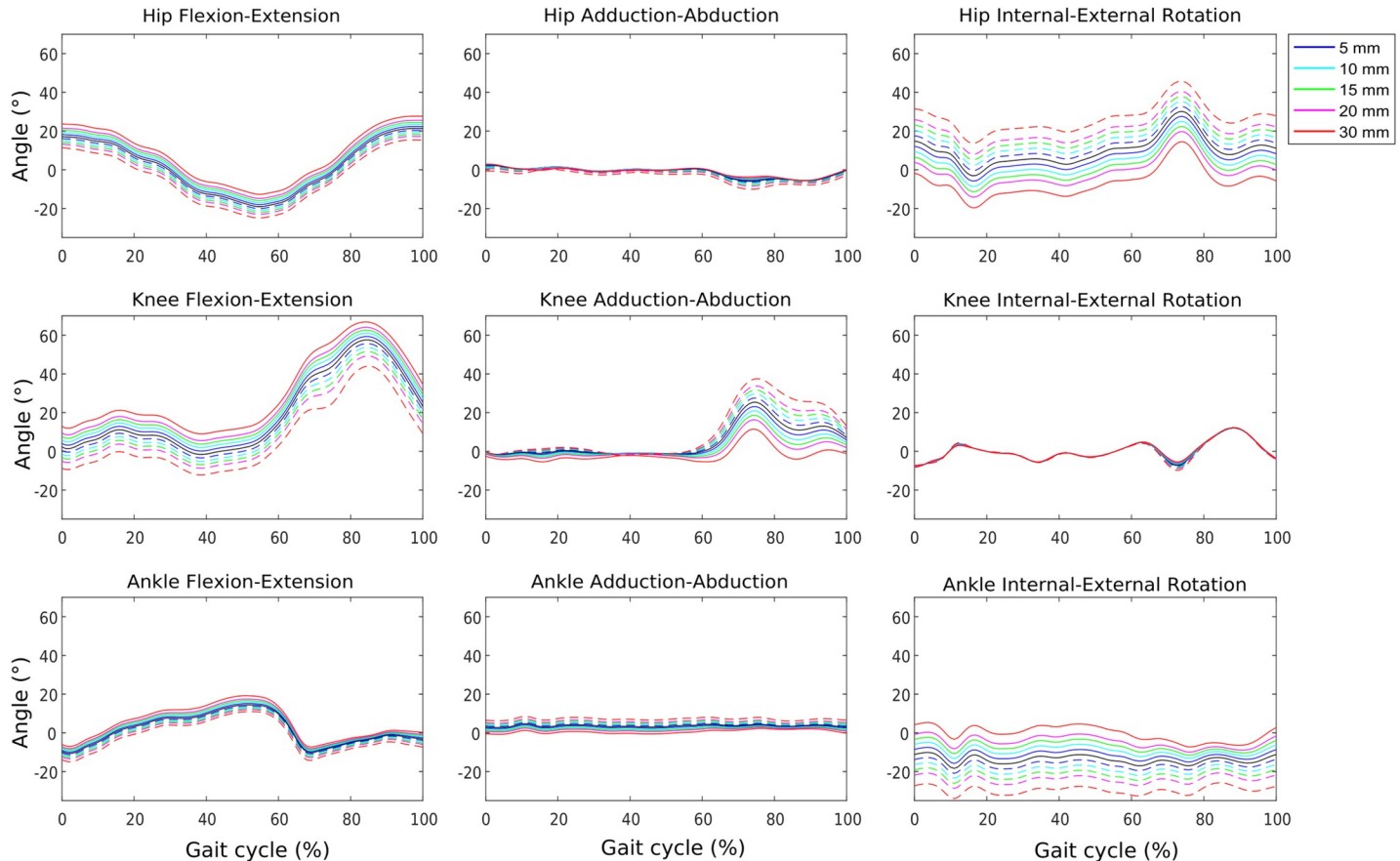

**Fig 3. KNE marker misplacement on anterior-posterior direction.** Kinematic deviations resultant from KNE marker misplacement on the anterior (solid lines) and posterior (dashed) directions at different magnitudes for one participant. One gait cycle is represented per condition.

rotation and the knee flexion-extension. These angles reached values over the 5° threshold for magnitudes of misplacement between 10 and 15 mm (Fig 2). Furthermore, knee adduction-abduction and hip flexion-extension were moderately affected as they presented an error over 5° only when the misplacement magnitude was above 20 mm. Finally, marker misplaced in

**Table 2. Correlation coefficients R between RMSD and magnitude of misplacement in percentage of leg length for both group.**

| Joint | Angle | Anterior | | Posterior | | Proximal | | Distal | |
|---|---|---|---|---|---|---|---|---|---|
| | | CP | TD | CP | TD | CP | TD | CP | TD |
| **Hip** | Flexion-Extension | 0.99 | 0.99 | 0.98 | 0.99 | 0.95 | 0.96 | 0.95 | 0.97 |
| | Adduction-Abduction | 0.75 | 0.77 | 0.54 | 0.67 | 0.78 | 0.74 | 0.75 | 0.76 |
| | Internal-External Rotation | 0.91 | 0.94 | 0.90 | 0.94 | 0.91 | 0.95 | 0.91 | 0.96 |
| **Knee** | Flexion-Extension | 0.98 | 0.96 | 0.97 | 0.95 | 0.97 | 0.97 | 0.91 | 0.97 |
| | Adduction-Abduction | 0.90 | 0.87 | 0.90 | 0.83 | 0.84 | 0.78 | 0.82 | 0.77 |
| | Internal-External Rotation | 0.58 | 0.61 | 0.59 | 0.71 | 0.83 | 0.93 | 0.92 | 0.96 |
| **Ankle** | Flexion-Extension | 0.96 | 0.85 | 0.94 | 0.84 | 0.94 | 0.88 | 0.86 | 0.78 |
| | Adduction-Abduction | 0.61 | 0.64 | 0.59 | 0.57 | 0.67 | 0.61 | 0.59 | 0.64 |
| | Internal-External Rotation | 0.91 | 0.89 | 0.92 | 0.90 | 0.94 | 0.94 | 0.83 | 0.82 |

All correlations resulted on a p-value <0.001. TD: Typically developed children.

the AP direction had a low influence on hip and ankle adduction-abduction, ankle flexion-extension and knee internal-external rotation as they did not present any error above 5˚ with any of the misplacements tested. Based on the results of Table 2, and being the maximal mean error obtained 5.5˚ (hip internal-external rotation) for a misplacement magnitude of 10 mm, we estimate that to ensure an error within 5˚ for all joint angles, a precision within approximately 1.2% of leg length on the AP direction needs to be assured. Thus, it is important to notice that a repeatability of marker placement of this magnitude might be difficult to obtain in clinical practice [7], especially for young children as they have a smaller morphology than adults.

A few major differences can be found between our results and the results reported by Osis et al. [18]. The major discordance is relative to the knee internal-external rotation where they reported an error of 5.1˚ with a misplacement of 10 mm in the AP direction. Our study showed only 0.7˚ for the same condition, which is in agreement with the small error observed in previous studies with the CGM markerset [14], [17]. More differences were found on hip abduction-adduction, knee flexion-extension, knee abduction-adduction and ankle internal-external rotation where Osis et al. [18] reported errors of 2.9˚, 1.6˚, 0.6˚ and <0.5˚ respectively while we obtained 0.6˚, 4.3˚, 3.5˚ and 4.5˚ respectively. The explanation for this difference may be because they analysed subjects while running and with a different marker set and biomechanical model which differs specially on the definition of the frontal plane of the thigh and shank segments [19]. Thus, the generalisation of the errors due to marker misplacement from one model to another is hazardous.

Kinematics were relatively unaffected by PD misplacement of the KNE marker as RMSD was generally presented to be under the 2˚ threshold, with the exception for the knee internal-external rotation that revealed a deviation up to 2.2˚ per 10 mm of misplacement. These results are in agreement with the results of Baker et al. [14] that presented more sensitivity of the knee internal-external rotation angle to PD misplacement than to AP misplacement (+0.4˚ and <0.1˚ respectively for a magnitude of 5 mm misplacement).

Deviations observed on internal-external rotation and flexion-extension angles are mainly characterized by offsets. Contrarily, the amplitudes of the adduction-abduction and internal-external rotation angles of the knee are mainly altered during the swing phase, where the knee is more flexed. Kadaba et al. (1990) reported similar observations. Such alterations correspond to the well-known cross-talk phenomenon [7] revealing a misorientation of the knee flexion axis (equal to the thigh medial-lateral axis).

The high sensitivity of the hip internal-external rotation angle to the misplacement of the KNE marker plays an important role in the clinical assessment of patients with CP. Indeed, this pathology frequently presents excessive femoral anteversion to which hip internal-external rotation angle is an indicator [24]. An error on this angle could lead to an erroneous evaluation that could potentially affect, for instance, the decision making for a derotation procedure [25]. Excessive hip external rotation and ankle internal rotation may be caused by anterior misplacement of KNE marker while excessive hip internal and ankle external rotation may be caused by a posterior misplacement (Fig 3). Furthermore, from our results, we can conclude that interpretation of gait data with KNE marker misplaced in the anterior and posterior directions may lead to a false interpretation of hyperflexion and hyperextension respectively on the sagittal plane.

The deviations due to marker misplacement have a very good positive correlation with the leg length, especially, hip internal-external rotation and all flexion-extension angles (R > 0.9). On the contrary, knee internal-external rotation and ankle adduction-abduction showed moderate correlation with magnitude of the misplacement expressed in percentage of leg length (R < 0.61). Those results show that a model addressing the effect of marker misplacement on

the kinematics could be developed and used in CGA to support the placement of the marker and the interpretation of data. The model presented in this study can estimate the kinematics for a misplacement considered in percentage of leg length and could be added to the report of a specific patient.

Various optimisation techniques were proposed in the past to reduce the impact of marker misplacement on the output kinematics [26–30]. Those methods were observed to reduce the effect of lack of marker placement precision and could be tested as a complement of the CGM in order to reduce the sensibility of this biomechanical model to marker misplacement [30]. For instance, the calibration method proposed by Baker et al. [26] aiming to compensate the mis-alignment of thigh markers could be implemented, serving as a checking for the placement of the markers and thus supporting the training of examiners. In addition, a well validated and standardized guidelines for the placement of the markers is crucial to produce in order to reduce the effects of incorrect anatomical landmark identification [20].

Additionally, the RMSD differences calculated between the CP and the TD group showed no significant difference except on the hip flexion-extension for misplacements in the AP and PD directions (Table 1) but the misplacements errors were of similar magnitude. The correlation between RMSD and magnitude of marker misplacement also did not present a significant difference between both groups (Table 2). We can conclude that the impact of KNE misplacement has a similar effect on gait kinematics in children who are typically developed and with CP ranging between GMFCS I and II. Nevertheless, we recommend to include both CP and the TD groups in the upcoming studies on the impact of other marker mispalcements or on the effect of the optimisation techniques proposed to reduce this impact.

Several limitations can be detected in the present study. The first is that the reference position of the KNE marker is dependent on palpation and incorporates the subjectivity associated with marker placement precision. To limit this effect, all marker placements were performed by a single experienced operator. However, as the results showed mainly an offset of the kinematic curves due to marker misplacement for every patient and that every patient is likely to have different physical misplacement errors, we conclude that this inherent error added to the reference position has a low impact on the results of this study. The second limitation is that the influence on the kinematics of only one marker misplacement was assessed when, in practice, the variability is the result of the combination of misplacements on the whole marker set. For instance, an erroneous definition of the femur orientation due to a misplacement of the KNE marker can be mitigated or intensified by a misplacement of other markers, such as thigh markers [7], [26] and pelvic markers [18]. Nonetheless, only the KNE marker was introduced in this study due to its central place in the CGM and for the clarity of the analysis. Further study will analyse the effect of the interactions of simultaneous misplaced markers. The third limitation is that different soft tissue artefacts were not considered among different positions of the marker which constitutes another difference with a real experimental misplacement. Moreover, with only kinematic data incorporated, this study lacks on information relative to the sensitivity of the kinetics of the CGM under marker misplacement conditions. Finally, the study used as reference for the uncertainty of marker placement [9] was based on a small healthy adult population. A similar study on a young population would be of great interest to scale the range of misplacements introduced in this study.

## Conclusion

This study demonstrated that the kinematics of the CGM is highly sensitive to KNE marker misplacement over the AP direction when performing gait analysis in children with CP. The most affected joint angles are the hip and ankle internal-external rotation, and the knee

flexion-extension. In order to obtain an accuracy below 5˚ for the hip rotation profile, the misplacement of the KNE marker in the AP direction should be lower than 1.2% of the leg length. In clinical gait analysis, error may considerably impact the management of motor disorders, especially when considering hip internal-external rotation profile. This study shows that a model of marker misplacement can be developed for the KNE marker. Indeed, its impact on kinematics is linear in function of the magnitude expressed in percentage of the leg length, apart for the knee internal-external rotation and ankle adduction-abduction. This study also showed that CGM is equally sensitive to misplacement of KNE the marker on typically developing children and children with CP (GMFCS I and II). Moreover, we can conclude that there is a potential for the development of a general model of marker misplacement to add a margin of errors on reports of CGA and support clinical decision.

## Supporting information

**S1 Table. Regression equation parameters.** Slope (m) and y-intercept (b) relatively to the regression equation of RMSD calculated and magnitude of misplacement expressed in terms of leg length.
(PDF)

**S1 Fig. Correlation between RMSD calculated error and magnitude of misplacement in percentage of leg length.** Scatter plot representing the RMSD for all tested magnitudes for misplacement of KNE marker in the AP direction considering the CP population. R (Correlation coefficient).
(PDF)

**S2 Fig. Prediction of kinematics based on misplacement magnitude from leg length.** Representation of prediction based on the regression equation described in S1 Fig and S1 Table for one patient and for a misplacement of the lateral epicondyle marker on anterior-posterior direction. Mean and standard deviation of the reference kinematics (blue line and shadow respectively). Predicted mean kinematics for a misplacement of 10mm on the anterior (red solid line) and posterior direction (red dashed line) directions.
(PDF)

## Author Contributions

**Conceptualization:** Mickael Fonseca.

**Data curation:** Mickael Fonseca.

**Formal analysis:** Stéphane Armand.

**Funding acquisition:** Stéphane Armand.

**Investigation:** Mickael Fonseca.

**Methodology:** Mickael Fonseca, Fabien Leboeuf.

**Software:** Mickael Fonseca, Fabien Leboeuf.

**Supervision:** Raphaël Dumas, Stéphane Armand.

**Validation:** Mickael Fonseca, Xavier Gasparutto, Raphaël Dumas, Stéphane Armand.

**Visualization:** Mickael Fonseca.

**Writing – original draft:** Mickael Fonseca.

**Writing – review & editing:** Mickael Fonseca, Xavier Gasparutto, Fabien Leboeuf, Raphaël Dumas, Stéphane Armand.

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
