## [Decision Letter · Decision Letter 0]

4 Dec 2019

PONE-D-19-30115

Impact of knee marker misplacement on gait kinematics of children with cerebral palsy using the Conventional Gait Model - A sensitivity study

PLOS ONE

Dear Mr. Fonseca,

Thank you for submitting your manuscript to PLOS ONE. After careful consideration, we feel that it has merit but does not fully meet PLOS ONE’s publication criteria as it currently stands. Therefore, we invite you to submit a revised version of the manuscript that addresses the points raised during the review process.

ACADEMIC EDITOR: Both reviewers present positive evaluations but have some major concern too especially the reviewer 1. Please consider their suggestions to improve the quality of manuscript.

We would appreciate receiving your revised manuscript by Jan 18 2020 11:59PM. To enhance the reproducibility of your results, we recommend that if applicable you deposit your laboratory protocols in protocols.io, where a protocol can be assigned its own identifier (DOI) such that it can be cited independently in the future. For instructions see: http://journals.plos.org/plosone/s/submission-guidelines#loc-laboratory-protocols

We look forward to receiving your revised manuscript.

Kind regards,

Kei Masani

Academic Editor

PLOS ONE

Journal Requirements:

3. Please upload a copy of Figure 4, to which you refer in your text on page 10. If the figure is no longer to be included as part of the submission please remove all reference to it within the text.

Reviewers' comments:

Reviewer's Responses to Questions

**Comments to the Author**

1. Is the manuscript technically sound, and do the data support the conclusions?

Reviewer #1: Yes

Reviewer #2: Yes

2. Has the statistical analysis been performed appropriately and rigorously? 

Reviewer #1: I Don't Know

Reviewer #2: Yes

3. Have the authors made all data underlying the findings in their manuscript fully available?

Reviewer #1: Yes

Reviewer #2: Yes

4. Is the manuscript presented in an intelligible fashion and written in standard English?

Reviewer #1: Yes

Reviewer #2: Yes

5. Review Comments to the Author

Reviewer #1: This manuscript examines the errors associated with misplacement of knee markers in CGS involving children with CP. Overall the manuscript is well written.

Major comments:

One of the main issues with the work relates to the clinical relevance. Specifically, the authors test misplacements from 5 - 30mm, and based on their results make recommendations about values that produce errors greater than the clinically accepted 5 degrees; however, it is unclear what magnitudes of misplacement are actually observed clinically. One citation provided in the introduction involves a small sample of adults, but this is not directly relevant to the population of interest here. Hence to allow for a more meaningful interpretation of the findings, the authors need to provide a stronger supported rationale and data about the reliability of marker placement at the knee.

It appears that previous work has established that AP and not PD displacements are primarily problematic towards achieving accurate kinematic measurements. Hence, this brings into question the novelty of this work in examining not only the magnitudes but directions. Please provide a stronger rationale for the need to examine the directions. Also, in the discussion, please clearly outline the new findings that this work has yielded.

The work would provide greater insights if it had included a population of age-matched typically developing children, to test the assumption that this type of work needs to involve a patient population. Furthermore, the level of impairment of the CP population needs to be provided (i.e. GMFCS level or other) in the methods.

More information needs to be provided about the data collection protocols (marker placement, filtering of data, etc). Who applied the markers? What training did they receive? How accurately were they able to apply the markers, and in particular the knee marker as its physical position could influence the findings? What assurances were made that the marker placements were correctly placed?

A major limitation as acknowledged by the authors is that only one knee marker was assessed. Hence the effects of the entire marker placement protocol can not be deduced. Given the protocol did not require actual physical misplacements as these were assessed virtually, it would be very feasible to explore the effects and interactions of misplacements of other markers. The authors should either add these analyses or provide a strong justification for their exclusion.

Minor comments:

The figures are of very poor quality and difficult to read.

Line 304 please check grammar

Line 307 please describe the calibration methods.

Reviewer #2: This manuscript presents a study that evaluated the impact of the misplacement of the lateral femoral epicondyle marker on the hip, knee and ankle joints kinematics. It also investigated if such an impact can be predicted as a function of the leg length. This study particularly targeted the the gait of children with CP. It was concluded that the lateral femoral epicondyle marker misplacement in the AP direction plays a significant role on the reliability of gait assessment.

The general research question has been an important topic since 30 years and the particular focus on children with CP is a novel contribution. The methodology is sound and te manuscript is very well-written and well structured. The manuscript is suitable for publication in PLOS ONE after addressing the following minor comments:

1- It is recommended to reduce the number of acronyms specially when they are not repeated through the manuscript several times.

2- Line 81, should be revised to “… [2].”

3- Is Eq 1 considered in the segment’s local frame or global frame? It should be clarified.

4- Line 188, “as: poor if R≤0.2, fair if 0.2< R ≤0.4,” the reviewer is not convinced if one should consider 0.2< R ≤0.4 as fair. How the classified is done in the literature other than [19]?

5- Line 196: should be revised to “…direction. Table 1…”

6- Line 208: Term ROM does not appear in the table. Why should it appear in the caption.

7- Table 1: what do bold fonts mean in the table?

8- Line 25: “5.48” degree: Does the number of significant digits correspond to the resolution of the measurements with cameras?

6. PLOS authors have the option to publish the peer review history of their article (what does this mean?). If published, this will include your full peer review and any attached files.

Reviewer #1: Yes: Jan Andrysek

Reviewer #2: Yes: Hossein Rouhani

---

## [Author Response · Author response to Decision Letter 0]

11 Mar 2020

To the reviewer #1:

We thank you for the remarks. They highlighted accurately the week points of our study.

There was an initial difficulty to highlight the novelty of the results provided by our study which were better described following your remarks.

Moreover, your suggestion to include a control group was followed and allowed us to provide information relative to the difference between both groups when adressing the impact of KNE marker misplacement. 

Therefore, your remark considering the fact that we used a study (Della Crocce 1999) as reference for the accuracy of marker placement on the lateral femural epicondyle marker higlighted not only the need for a similar study providing results on different population (young population) but also helped us to clarify this limitation on our study.

To the reviewer #2

We thank you for the remarks to our study. They were useful to improve the manuscript and to correct incorrections.

---

## [Decision Letter · Decision Letter 1]

24 Mar 2020

PONE-D-19-30115R1

Impact of knee marker misplacement on gait kinematics of children with cerebral palsy using the Conventional Gait Model - A sensitivity study

PLOS ONE

Dear Mr. Fonseca,

Thank you for submitting your manuscript to PLOS ONE. After careful consideration, we feel that it has merit but does not fully meet PLOS ONE’s publication criteria as it currently stands. Therefore, we invite you to submit a revised version of the manuscript that addresses the points raised during the review process.

ACADEMIC EDITOR: Please follow the reviewer 1's comments to further improve the readability of manuscript.

We would appreciate receiving your revised manuscript by May 08 2020 11:59PM. To enhance the reproducibility of your results, we recommend that if applicable you deposit your laboratory protocols in protocols.io, where a protocol can be assigned its own identifier (DOI) such that it can be cited independently in the future. For instructions see: http://journals.plos.org/plosone/s/submission-guidelines#loc-laboratory-protocols

We look forward to receiving your revised manuscript.

Kind regards,

Kei Masani

Academic Editor

PLOS ONE

Reviewers' comments:

Reviewer's Responses to Questions

**Comments to the Author**

1. If the authors have adequately addressed your comments raised in a previous round of review and you feel that this manuscript is now acceptable for publication, you may indicate that here to bypass the “Comments to the Author” section, enter your conflict of interest statement in the “Confidential to Editor” section, and submit your "Accept" recommendation.

Reviewer #1: (No Response)

Reviewer #2: All comments have been addressed

2. Is the manuscript technically sound, and do the data support the conclusions?

Reviewer #1: Yes

Reviewer #2: Yes

3. Has the statistical analysis been performed appropriately and rigorously? 

Reviewer #1: Yes

Reviewer #2: Yes

4. Have the authors made all data underlying the findings in their manuscript fully available?

Reviewer #1: Yes

Reviewer #2: Yes

5. Is the manuscript presented in an intelligible fashion and written in standard English?

Reviewer #1: Yes

Reviewer #2: Yes

6. Review Comments to the Author

Reviewer #1: Overall, the authors have done a good job to address the issues.

There are some minor issues remaining. Also, the paper should be grammatically improved. Some specifics are noted below.

Line 34 – Correct grammar to ‘The lower limb kinematics were calculated for each misplacement and the resultant outputs were compared with the original..’

Line 38 , grammar – change ‘is’ to ‘are’

Lime 58 grammar – change ‘on’ to ‘in’

Line 161 and 163 – Should say ‘as a function of’

Line 190 and elsewhere – Change ‘On’ to ‘In’ when referring to the direction

Line 208 – change ‘in’ to ‘on’

Line 215 – remover the word ‘a’ or add the word ‘of’

Line 242 and some other places, the sentences are confusing

Specifically the sentence is ‘Our results demonstrated that the largest error on the kinematics introduced by the KNE marker misplacement was in the AP direction ’ and I believe it should say something like ‘Our results demonstrated that the largest errors in the kinematics occurred when the KNE marker was misplaced in the AP direction..’

I appreciate the authors including the data on the TD children. In the discussion it might be appropriate to suggest the implications of the findings. One potential suggestion is that perhaps some of the future research (such as examining the other markers) could be done on TD populations and generalized to patient populations to reduce the burden on the latter.

Quality/resolution of the figures need improvement

Table 1 – define what the asterisk represents. Also clarify in the captions why peak values were not entered and compared for the TD group, or otherwise enter those values

Table 2 – please consider adding the correlation results for the TD kids, as this is relevant.

Reviewer #2: The manuscript is revised and the reviewer's comments are properly addressed. The manuscript is suitable for publication in PLOS ONE.

7. PLOS authors have the option to publish the peer review history of their article (what does this mean?). If published, this will include your full peer review and any attached files.

Reviewer #1: Yes: Jan Andrysek

Reviewer #2: Yes: Hossein Rouhani

---

## [Author Response · Author response to Decision Letter 1]

27 Mar 2020

We thank reviewer 1 for the comments regarding the grammatical aspect. It was taking into in consideration and the manuscript was improved. Moreover the advices to include the missing data of TD group also completed the results.

We thank the reviewer 2 for the revision of the manuscript.

---

## [Decision Letter · Decision Letter 2]

7 Apr 2020

Impact of knee marker misplacement on gait kinematics of children with cerebral palsy using the Conventional Gait Model - A sensitivity study

PONE-D-19-30115R2

Dear Dr. Fonseca,

We are pleased to inform you that your manuscript has been judged scientifically suitable for publication and will be formally accepted for publication once it complies with all outstanding technical requirements.

With kind regards,

Kei Masani

Academic Editor

PLOS ONE

Additional Editor Comments (optional):

Reviewers' comments:

Reviewer's Responses to Questions

**Comments to the Author**

1. If the authors have adequately addressed your comments raised in a previous round of review and you feel that this manuscript is now acceptable for publication, you may indicate that here to bypass the “Comments to the Author” section, enter your conflict of interest statement in the “Confidential to Editor” section, and submit your "Accept" recommendation.

Reviewer #1: All comments have been addressed

2. Is the manuscript technically sound, and do the data support the conclusions?

Reviewer #1: Yes

3. Has the statistical analysis been performed appropriately and rigorously? 

Reviewer #1: Yes

4. Have the authors made all data underlying the findings in their manuscript fully available?

Reviewer #1: (No Response)

5. Is the manuscript presented in an intelligible fashion and written in standard English?

Reviewer #1: Yes

6. Review Comments to the Author

Reviewer #1: (No Response)

7. PLOS authors have the option to publish the peer review history of their article (what does this mean?). If published, this will include your full peer review and any attached files.

Reviewer #1: Yes: Jan Andrysek

---

## [Editor Report · Acceptance letter]

15 Apr 2020

PONE-D-19-30115R2 

Impact of knee marker misplacement on gait kinematics of children with cerebral palsy using the Conventional Gait Model - A sensitivity study 

Dear Dr. Fonseca:

I am pleased to inform you that your manuscript has been deemed suitable for publication in PLOS ONE. Congratulations! Your manuscript is now with our production department. 

With kind regards,

on behalf of

Dr. Kei Masani 

Academic Editor

PLOS ONE